# Development and Validation of a Fast and Sensitive UPLC-MS/MS Method for Ethyl Glucuronide (EtG) in Hair, Application to Real Cases and Comparison with Carbohydrate-Deficient Transferrin (CDT) in Serum

**DOI:** 10.3390/ijms26031344

**Published:** 2025-02-05

**Authors:** Leonardo Romani, Giulio Mannocchi, Federico Mineo, Francesca Vernich, Lucrezia Stefani, Luigi Tonino Marsella, Roberta Tittarelli

**Affiliations:** 1Laboratory of Forensic Toxicology, Section of Legal Medicine, Social Security and Forensic Toxicology, Department of Biomedicine and Prevention, Faculty of Medicine and Surgery, University of Rome “Tor Vergata”, Via Montpellier 1, 00133 Rome, Italy; leonardo.romani.09@students.uniroma2.eu (L.R.); f.mineo@med.uniroma2.it (F.M.); francesca.vernich@uniroma2.it (F.V.); lucrezia.stefani@students.uniroma2.eu (L.S.); marsella@uniroma2.it (L.T.M.); roberta.tittarelli@uniroma2.it (R.T.); 2PhD School in Medical-Surgical Applied Sciences, University of Rome “Tor Vergata”, Via Montpellier 1, 00133 Rome, Italy

**Keywords:** ethyl glucuronide (EtG), carbohydrate-deficient transferrin (CDT), alcohol abuse, ultra-performance liquid chromatography tandem mass spectrometry (UPLC-MS/MS), driving under the influence of alcohol, hair analysis, full validation

## Abstract

Alcohol is responsible for an ever-increasing number of deaths worldwide, and many road accidents are caused by irresponsible drinking and driving. The use of biomarkers that can support a diagnosis of alcohol abuse is a very important tool that can improve the prevention of many alcohol-related diseases and serious traffic accidents. The main aim of our study was the full validation of a rapid and simple method by ultra-performance liquid chromatography tandem mass spectrometry (UPLC-MS/MS) to detect ethyl glucuronide in hair (hEtG). The method was successfully applied to n = 171 real hair samples collected from drivers convicted of driving while impaired by alcohol or drugs. A comparison of hEtG and serum Carbohydrate-Deficient Transferrin percentages (% CDT) was also performed to carefully evaluate the data in relation to the specific detection windows of the two different biomarkers. Most of the drivers with hEtG > 30 pg/mg were males in their thirties. None of the hEtG-positives had a serum % CDT above the cutoff (≥2%). Although some researchers suggest caution until solid data are available on the possible effects of interindividual variability that may influence EtG incorporation and metabolism, hEtG is a very useful biomarker of long-term alcohol exposure that shows greater reliability than traditional blood markers.

## 1. Introduction

According to the World Health Organization (WHO) in 2019, about 2.6 million deaths worldwide were attributable to alcohol consumption. Of these, 2 million were men and 0.6 million were women. Although from 2010 to 2019 the number of alcohol-attributable deaths decreased by 20.2% globally, in 2019, the highest levels of alcohol-related deaths were observed in the European Union (EU) and African regions, with 52.9 and 52.2 deaths per 100,000 people, respectively [1]. Furthermore, in 2019, more than 150,000 deaths were caused by injuries attributable to alcohol, such as traffic accidents, falls, drownings, burns, assaults, sexual assault and suicides [1].

Ethanol consumption may have several clinical sequelae (liver and heart diseases, cancer, depression and anxiety) and forensic implications (occupational safety, suspension of driving licenses and firearm licenses, and child custody).

Ethanol biomarkers are important tools to discriminate abstinence from repeated or excessive alcohol intake, providing crucial information on alcohol consumption.

They can be divided into two categories: indirect and direct biomarkers. Indirect biomarkers are mainly used to detect alcohol abuse (i.e., longer-term intake), whereas direct biomarkers are used to measure recent intake [2]. Indirect biomarkers help clinicians detect the effects of alcohol both systemically, by aiding assessments of damage to organs and tissues, and biochemically through the alteration of body chemistry [3]. Some of indirect biomarkers are enzymes [e.g., alanine aminotransferase (ALT), aspartate aminotransferase (AST) and gamma glutamyl transpeptidase (GGT)] or proteins [e.g., carbohydrate-deficient transferrin (CDT) and cholesteryl ester transfer protein (CETP)] that undergo specific changes in response to acute or chronic alcohol consumption [4,5].

The direct biomarkers of ethanol intake are products of ethanol metabolism (e.g., ethanol itself, acetaldehyde and other metabolites). The most widely used direct biomarkers produced during non-oxidative alcohol metabolism are ethyl glucuronide (EtG) and ethyl sulphate (EtS).

EtG is a stable, water-soluble metabolite of ethanol produced enzymatically by the activity of UDP-glucuronosyltransferase (UGT) through the process of ethanol glucuronidation [6] (Figure 1).

In recent years, EtG has been considered the most reliable marker for assessing abstinence (which means no intake of any alcoholic beverages or other alcohol-containing products over a predefined period), and to distinguish occasional alcohol intake from chronic excessive use, defined as the average consumption of 60 g or more of pure ethanol per day over several months [7].

EtG can be identified in several biological matrices, such as blood and urine (to assess recent or short-term alcohol intake; it can be present up to 18 h in blood and up to 2-4 days in urine after heavy drinking) [8,9,10] and also hair (to evaluate long-term excessive alcohol use in a wider detection window) [11].

The Society of Hair Testing (SoHT) established that a concentration of EtG ≤ 5 pg/mg in the proximal head segment with a length of 3 cm to 6 cm is compatible with self-reported abstinence, whereas a concentration > 5 pg/mg refers to repeated alcohol consumption. EtG values ≥ 30 pg/mg strongly suggest chronic excessive alcohol consumption [7].

As reported in the literature, to perform EtG analysis on hair (hEtG), after different washing steps, the samples (weighing between approximately 15 and 200 mg) [12] must be shredded with clean scissors to obtain segments of about 0.1–0.3 cm in length, or the samples can be crushed by a ball mill [6,13,14].

The extraction procedure is usually performed with distilled water through simple incubation in an ultrasonic bath, or a combination of both methods. Distilled water proved to be a better solvent than methanol or a water/methanol mixture [6].

The international cutoff values recommended by the SoHT are defined according to the conditions described above (hair pulverization with water extraction). If different methodologies are used, it must be demonstrated that comparable results are obtained through proficiency testing [7].

A recent study by Salomone et al. showed that out of 709 samples analyzed, the average increase in concentration from cut to pulverised hair was 62.1%. Furthermore, 3.7% of the samples were positive if milled, but negative if cut with scissors. In both cases, hydrolysis was performed with water–methanol overnight. After incubation, the samples were sonicated [15].

Vincenti et al. proposed a fast pretreatment of the cut hair sample in a methanol–water mixture using an automated pressurized liquid extraction (PLE) system and subsequent solid-phase micro extraction (SPE), with significantly increased costs per analysis [16].

Groff et al. compared the overnight pretreatment of pulverised and cut hair in water, as suggested by the SoHT, with pretreatment in M3^®^ for 1 h at 100 °C. The best extraction was obtained by hair pulverization in water, which was totally comparable to that in M3^®^ with the hair cut with scissors [17].

In consideration of these important findings in the literature, a fast and low-cost method to detect hEtG using ultra-performance liquid chromatography tandem mass spectrometry (UPLC-MS/MS), with a pretreatment comparable to that suggested by the SoHT with the hair cut with scissors, was fully validated using buffer M3^®^.

The method was then successfully applied to n = 171 real hair samples collected from 1 October to 31 December 2024 at the Laboratory of Forensic Toxicology, University of Rome “Tor Vergata”, from drivers convicted of driving under the influence (DUI) of alcohol or drugs who applied to regain their driving license.

The period of observation was closely related to the introduction of hEtG analysis by the Local Medical Commissions, which are responsible for evaluating the mandatory medical examination to certify the mental and physical fitness of applicants in driver’s license regranting procedures.

A comparison of hEtG and serum CDT percentages (% CDT) was also performed to rigorously interpret the results and to carefully evaluate the data in relation to the specific detection windows of the two different biomarkers.

## 2. Results

A total of n = 171 real hair samples were analyzed using the UPLC-MS/MS method. The cutoff for an hEtG positive test was set at 30 pg/mg, as indicated by the SoHT. Among these hair samples, n = 20 samples were classified as positive (EtG ≥ 30 pg/mg) and n = 11 were within the range 21–29 pg/mg.

Approximately 30% of the samples (n = 51) had an EtG concentration between 5 and 20 pg/mg, while 52% (n = 89) had EtG values below 5 pg/mg (Figure 2).

A descriptive statistical approach for age and sex was used to collect and describe our data, with the aim to identify a trend of alcohol abuse in the studied population.

Most of the drivers tested for hEtG were men (n = 141; 82.5%), and n = 30 were women (17.5%). The mean age of the population was 29.9 years (a minimum mean age of 18 and a maximum mean age of 35), with a standard deviation of 4.36.

Among the 20 positive subjects, n = 16 were men (80%), while only n = 4 were women (20.0%). The average age of the people who tested positive was 30.65 (a minimum mean age of 21 and a maximum mean age of 35), with a standard deviation of 3.70. In positive cases, hEtG concentrations ranged from a minimum of 31 pg/mg to a maximum of 240 pg/mg, with corresponding quartiles (first quartile, median quartile and third quartile) of 34, 41 and 86 pg/mg, respectively. The mismatch between these values can be explained by the average daily ethanol consumption, which can vary depending on the subject.

There was no significant association between hEtG values and serum % CDT, as none of the samples were positive for both markers. There were only n = 2 blood samples out of the total collected (n = 171) that tested positive for CDT (with values > 2.0%), but in both cases, hEtG values were lower than 30 pg/mg.

There was also no biochemical evidence of liver damage induced by heavy drinking, as the indirect alcohol biomarkers ALT, AST and GGT were within the reference range in all the samples analyzed.

## 3. Discussion

One of the aims of our study was to compare two biomarkers of alcohol consumption, hEtG and CDT, to evaluate their reliability and efficiency in diagnosing chronic alcohol abuse in cases of forensic interest. For this reason, a fast and simple UPLC-MS/MS method for the determination and quantitation of hEtG was developed and fully validated according to the guidelines of the SoHT in terms of extraction and performance.

The use of the M3^®^ reagent led to a shorter pretreatment of the samples, making it suitable for high-throughput analysis.

The most interesting result was the disagreement between hEtG values and serum % CDT. None of the hEtG-positive drivers (n = 20 out of n = 171 samples) had a serum % CDT above the cutoff.

Furthermore, the two only subjects with a positive % CDT had hEtG values below the cutoff of 30 pg/mg.

These results can be explained not only by the reliability of the two biomarkers but also by their kinetics and different detection times.

The analysis of hEtG was performed on the proximal 3 cm of hair, so the results refer to alcohol consumption in the last 3 months (1 cm/month is the average hair growth rate), while CDT levels, which increase after alcohol consumption of 40–60 g/day for at least two weeks, with a half-life of about 10 days [18], refer to an alcohol intake in the period of approximately 14 days prior to blood collection.

Since CDT levels usually normalize in about 10 days and this information is now also readily available through the Internet, individuals who must undergo toxicological tests can discontinue alcohol use in the weeks prior to blood collection. This kind of behavior makes the identification of heavy alcohol users particularly difficult.

Although CDT is widely used as a biomarker for the diagnosis of chronic alcohol abuse [19], it has been shown to have limited sensitivity [20]. Caution is needed in the interpretation of the data, especially in pregnant women and patients with high body mass index (BMI), diseases of the liver and biliary system, iron metabolism disorders, cancer, estrogen therapy, transplantation, antiepileptic drugs, genetic variants (GVs), carbohydrate-deficient glycoprotein syndromes (CDG I and II) and rare hereditary glycoprotein disorders [5,21,22,23,24].

Conversely, EtG, being a direct metabolite of ethanol, shows a higher reliability [25,26], and it is not affected by other factors. Hair color, sex, age, BMI, smoking and cosmetic treatment do not seem to influence hair analysis for EtG [27] (Table 1).

EtG is very stable in hair [28] and enables the monitoring of long-term alcohol consumption habits as well as abstinence. It can also be very useful as a biomarker in other matrices (e.g., urine) for all the conditions that require so-called zero tolerance alcohol limits (e.g., novice drivers who have been licensed for less than 3 years, young drivers up to 21 years old and professional drivers) [29].

Although some researchers suggest caution until solid data are available on the possible effects of interindividual variability and other confounding factors [30,31,32] different from alcohol that may influence EtG incorporation and metabolism, hEtG is becoming a very useful biomarker of long-term alcohol exposure that shows greater reliability than traditional blood markers.

Our study has some limitations: the study population consisted of licensed drivers over 18 years old, and the number of samples analyzed was limited because only in the last few months, the Local Medical Commissions have introduced EtG as a biomarker of alcohol abuse. However, the data collected suggest that further investigations on this topic are warranted, potentially involving the application of different biomarkers for better discriminating the recreational use of alcohol from an excessive chronic alcohol consumption. Further studies will be carried out to better assess the impact of alcohol on driving ability by evaluating their prevalence in alcohol- and drug-impaired road accidents.

## 4. Materials and Methods

### 4.1. Sample Collection

Hair and blood samples were collected under a rigorous chain-of-custody process at the Laboratory of Forensic Toxicology, University of Rome “Tor Vergata”, from drivers convicted of impaired driving. The tests were mandatory as required by the Local Medical Commission (LMC) for fitness to drive, and analyses were performed six months after driver conviction. These data, collected for nonmedical purposes, were aggregated and anonymized before evaluation.

The inclusion criteria were that all the selected people were drivers aged older than 18 years with a suspended license due to driving under the influence of drugs or alcohol. Subjects under the age of 18 years or those who attended our laboratory for toxicological analysis for other purposes (e.g., firearm license, international adoption, legal separations and child custody) were excluded from the study.

The sample collection and the analyses were carried out in agreement with the guidelines of the Society of Hair Testing (SoHT) and the Italian Group of Forensic Toxicologists (GTFI) [33,34].

Hair samples were collected as close to the scalp as possible from the posterior region of the head. The strands, thick as a pencil, were tied with a thread at the proximal end to distinguish their alignment and were kept in an envelope in a dry and dark place at room temperature until analysis. The head hair was divided into two aliquots (aliquots A and B). No less than 100 mg for each aliquot was collected to perform both the qualitative and quantitative analyses (aliquot A), and another sample was retained for a possible counter-analysis (aliquot B). The analyses were performed on the first 4 cm of scalp hair to investigate the period of 3-4 months prior to the hair collection.

Alternatively, as stated in the guidelines, about 200 mg of hair was collected from other body parts (chest, arm and legs). Axillary hair was not suitable as an alternative matrix [35].

For CDT analysis, the blood samples were collected using Vacutainer tubes with a separating gel and coagulation activator. After centrifugation, 100 μL of serum was transferred to a 1.5 mL Nerbe Plus microcentrifuge tube (Nerbe Plus GmbH & Co. KG Porschestraße 25, 21423 Winsen/Luhe Deutschland), and the residual was stored at −20 °C. Additional tests were also performed to highlight any possible alteration in liver enzymes (ALT, AST and GGT) and/or a change in mean corpuscular red blood cell volume (MCV) [36].

### 4.2. HPLC Analysis for CDT Determination

In our study, the Eureka srl Lab Division kit (Eureka Lab Division, Sentinel Diagnostics, Chiaravalle, 60033, Italy) was used [37], and a % CDT equal to or higher than 2% was considered positive [38]. The % CDT was calculated as the percentage of asialo–monosialo–disialo transferrin in relation to the total transferrin [39,40] using the baseline integration mode (Figure 3), as suggested by the working group of the International Federation of Clinical Chemistry (IFCC).

The method for CDT analysis was previously described by Fiorelli et al. [5].

### 4.3. UPLC-MS/MS for hEtG Analysis

#### 4.3.1. Chemicals and Reagents

The stock standard solutions of EtG (Ethyl-Beta-D-glucuronide) along with deuterium-labeled stock solution EtG d5 (Ethyl-Beta-D-glucuronide-D5) that were used as internal standards (ISs) were supplied by LGC Standards (LGC Standards S.r.l. Sesto San Giovanni, Milan, Italy).

LC-MS-grade water and methanol were purchased from Sigma-Aldrich^®^ (Milan, Italy).

The M3^®^ reagent used for hair extractions and the liquid chromatography mobile phase A (based on water) and mobile phase B (based on organic solvent) were supplied by Comedical^®^ s.r.l. (Trento, Italy).

#### 4.3.2. Preparation of Calibration and Quality Control Samples

Two different working solutions (0.05 µg/mL and 0.5 µg/mL) that ranged from limit of quantification (LOQ = 2) to 300 pg per mg hair were prepared by diluting stock standard solution in methanol. An internal standard solution containing 0.5 µg/mL EtG d5 was prepared by diluting an EtG deuterium-labeled solution with methanol. Low-, medium- and high-quality control (QC) working solutions were prepared daily at three required concentrations spanning the linear dynamic range of the calibration curve. Stock, working and internal standard solutions were all stored at −20 °C until analysis.

#### 4.3.3. EtG-Free Hair Samples

EtG-free hair samples were donated from laboratory personnel. Hair collected from alcohol abstinent volunteers were naturally colored and cosmetically treated (bleached and dyed). Thirty real samples were gently donated as discarded material by the Section of Legal Medicine, Social Security and Forensic Toxicology, Department of Biomedicine and Prevention, University of Rome Tor Vergata, and stored at room temperature in a dry and dark place.

#### 4.3.4. Sample Preparation

Hair samples were washed three times with water during a decontamination process and finely cut. Aliquots of 25 mg were prepared in glass tubes, the proper amounts of working standard/IS solutions and 500 µL of M3^®^ reagent were added, and the aliquots were incubated at 100 °C for 60 min. Samples were cooled at room temperature and then transferred to autosampler vials (Figure 4).

#### 4.3.5. Instrumentation

Analysis was performed using a UPLC Acquity I Class (Waters, Milford, MA, USA) equipped with a Waters Atlantis™ Premier BEH C18 AX Van Guard™ FIT (2.1 × 100 mm and 2.5 μm particle size) Column 1/pk with a Waters Van Guard Column Protection™ precolumn kept at 50 °C. The chromatographic system was interfaced to a Waters XEVO TQD (Waters, Milford, MA, USA) tandem quadrupole mass spectrometer. The chromatographic run lasted for 7 min, with a gradient mobile phase composed of mobile phase A (aqueous solution) and mobile phase B (organic solvent) belonging to the Comedical M3^®^ Line, at a flow rate 0.4 mL/min. Initial conditions were 98:2 (A/B) up to 0.50 min. Subsequently, the gradient trend was as follows: 2.90 min 50:50 (A/B), from 3 min to 4.5 min 0/100 (A/B). From 4.5 min, the gradient came back to the initial conditions, 98:2 (A/B) by 8.0 min. Mass spectrometric analysis was performed in negative ion multiple reaction monitoring (ES-MRM) mode. The capillary voltage was 0.5 kV, cone was 50 V, desolvation temperature was 650 °C and desolvation gas flow was 1200 L/h. Two transitions for EtG and one transition for EtG d5 were selected (Figure 5). Data were collected in the Waters TargetLynx XS V. 4.2 SCN1045 and were processed with Microsoft Excel^®^ 2016 MSO (16.0.4738.1000) (Microsoft Corporation^®^, Redmond, WA, USA).

Transitions, relative cone voltage (V) and collision energy (CE) are reported in Table 2.

### 4.4. Method Validation

The developed method was validated according to updated international guidelines in the field of our research [41,42]. The following parameters were evaluated for a quantitative method: selectivity, carryover, linearity, accuracy, precision, limit of detection (LOD), low limit of quantification (LLOQ), matrix effects, recovery and dilution integrity.

#### 4.4.1. Selectivity

A total of 50 blank samples fortified with several analytes including pharmaceuticals (benzodiazepines, fentanyl, oxycodone and tramadol) and drugs of abuse (cocaine, opioids, methadone, amphetamines and cannabinoids), but not EtG, were tested and evaluated along with two blank samples fortified with only internal standards (zero samples). Moreover, EtG-free hair samples (ten different sources) were analyzed to evaluate possible matrix interferences with analytes.

#### 4.4.2. Carryover

A blank sample was injected after the highest concentration calibrator for an evaluation of the carryover signal. The signal must be lower than the method’s LOD. The results were confirmed using triplicate EtG analysis.

#### 4.4.3. Linearity

Linearity was evaluated after injecting six different daily replicates of calibration points (LOQ; 5 pg/mg; 15 pg/mg; 30 pg/mg; 90 pg/mg; and 300 pg/mg) over four subsequent working days.

#### 4.4.4. Accuracy, Precision, Limit of Detection (LOD) and Low Limit of Quantification (LLOQ)

Intra-day and inter-day precision and accuracy were evaluated by analyzing three different QC samples in five replicates (low QC = 6.0 pg/mg, medium QC = 25 pg/mg and high QC = 240.0 pg/mg). Intra-day and inter-day precision were considered acceptable when lower than 15% (CV %) while bias is between ±15%. Limits of detection (LODs) and LLOQ were evaluated with spiked samples with decreasing concentrations of analyte.

Accuracy, precision, limit of detection (LOD) and low limit of quantification (LLOQ) are reported in Table 3.

#### 4.4.5. Matrix Effect, Process Efficiency and Recovery

These parameters were determined according to Matuszewski et al. [42]. Matrix effect, recovery and process efficiency are reported in Table 3.

#### 4.4.6. Dilution Integrity

Dilution integrity was tested with blank samples fortified with analytes of interest whose concentrations were 5 and 10 times higher than the highest calibration point to verify that precision and accuracy were within 15%.

## 5. Conclusions

EtG in hair is a reliable and functional marker in the diagnosis of alcohol abuse. The development and validation of a fast, sensitive and specific UPLC-MS/MS method was a good starting point for a proper technical approach for the determination and quantification of EtG. A crucial step was the hair sample preparation.

The SoHT consensus reports that the main sample pretreatment is hair pulverization for EtG extraction in water. Groff et al. [17] confirmed that the recovery obtained by hydrolysis in M3^®^ (1 h at 100 °C) is comparable to hydrolysis for hair pulverized in water (overnight at room temperature and subsequent sonication).

Therefore, the validation of a method using M3^®^ reagent allowed us to reduce the pretreatment time of samples, making them suitable for high-throughput analysis and maintaining the standards required by the SoHT.

Furthermore, comparisons with % CDT should be evaluated with caution to diagnose alcohol abuse. Alcohol consumption in both clinical and forensic settings should be carefully assessed using several biomarkers that can cover different windows of detection and allow for a more in-depth analysis of each case.

In this context, our method is a good compromise among those reported in the literature for time of analysis, costs and cost-effectiveness, making it widely applicable in clinical settings for monitoring alcohol abstinence and as a support in the multidisciplinary approach to excessive and chronic alcohol consumption.

Our method will be applied in forensics to a larger population of samples to investigate possible factors that may interfere with the determination and quantification of EtG in hair. Future goals also aim to introduce new markers of alcohol abuse that may be useful in the diagnosis of alcohol abuse in the forensic setting.

## Figures and Tables

**Figure 1 ijms-26-01344-f001:**
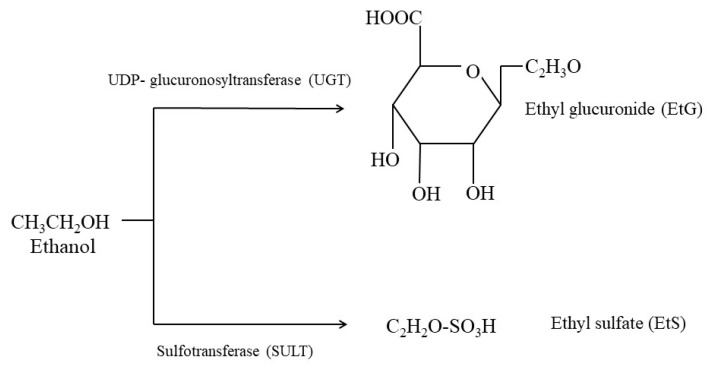
Production of ethyl glucuronide and ethyl sulphate via UDP-glucuronosyltransferase (UGT) and sulphotransferase (SULT), respectively.

**Figure 2 ijms-26-01344-f002:**
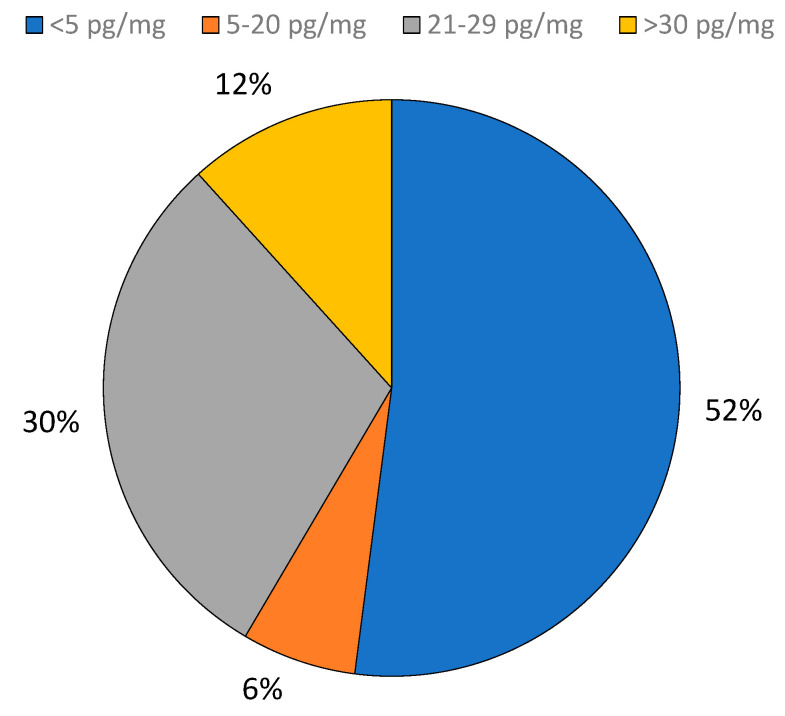
Concentration of hEtG in the real hair samples analyzed with the UPLC-MS/MS method.

**Figure 3 ijms-26-01344-f003:**
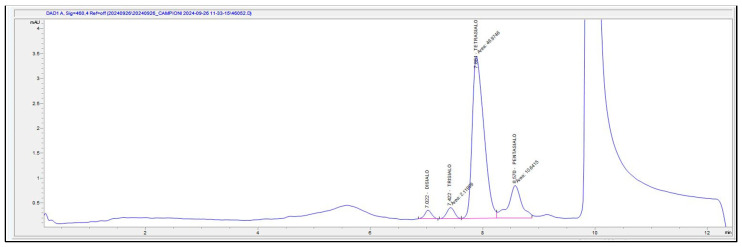
HPLC-DAD chromatogram (460 nm) of serum with an abnormal CDT level (2.30%).

**Figure 4 ijms-26-01344-f004:**
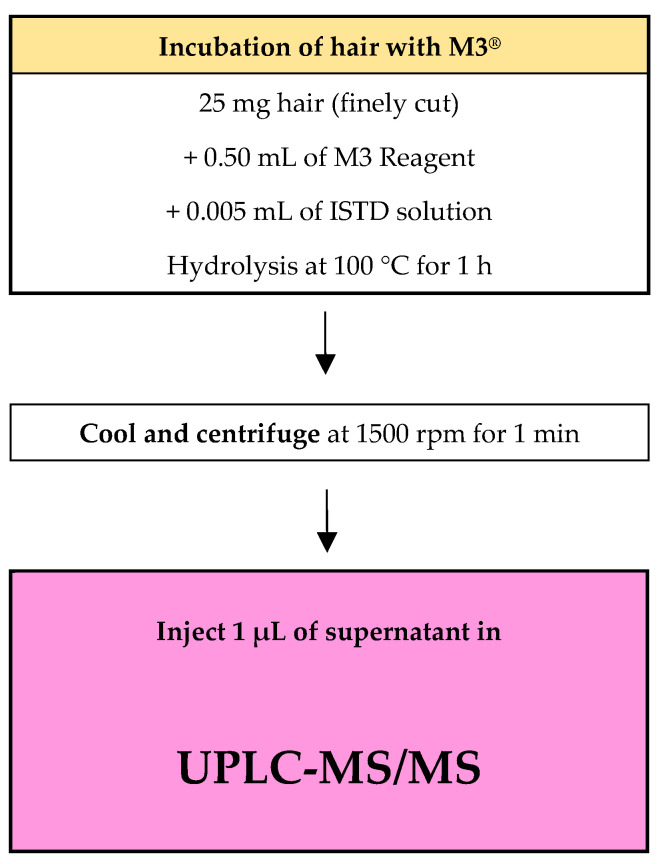
Workflow approach for sample preparation.

**Figure 5 ijms-26-01344-f005:**
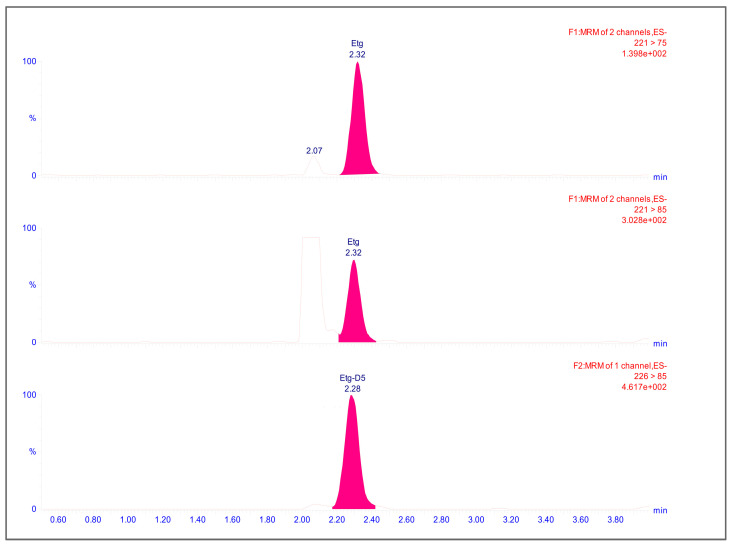
UPLC-MS/MS chromatogram of EtG real hair sample at a concentration of 40 pg/mg.

**Table 1 ijms-26-01344-t001:** Comparison between hEtG and CDT as biomarkers of alcohol abuse.

	hEtG	CDT
Type of marker	Direct	Indirect
Interferences from health conditions	No	Yes (Liver disease, pregnancy, genetic variants, cancer)
Window of detection	About 3 months	7-14 days
Analytical technique	UPLC/MS-MS	HPLC-DAD
Cost-effectiveness	High costs/the most effective tool for long-term detection of excessive use of alcohol	Low costs/reduced informativeness over the medium to long term
Reliability	High sensitivity and specificity	High specificity, but a limited sensitivity
Availability	Noninvasive sampling, low infectious risk	Invasive sample, high infectious risk

**Table 2 ijms-26-01344-t002:** Retention times (RTs) and mass spectrometry parameters for all target compounds.

RT (min)	MRM Transition				Cone Voltage (V)	Compound
	CE (eV)	Qualifier (*m*/*z*)	CE (eV)	Quantifier (*m*/*z*)		
2.32	17.0	221→85	15.0	221→75	**35**	**EtG**
2.28	-	-	17.0	226→85	35	EtG D_5_

Legend: CE, collision energy.

**Table 3 ijms-26-01344-t003:** Validation parameters for EtG.

Compound	Determination Coefficient (r^2^)	LOD (pg/mg)	LLOQ (pg/mg)	Accuracy (% Error) QC			Intra-Assay Precision (% CV) QC			Inter-Assay Precision (% CV) QC			Matrix Effect (%)	Recovery (%)	Process Efficiency (%)
				Low	Mid	High	Low	Mid	High	Low	Mid	High			
EtG	0.998 ± 0.003	0.75	2.5	5.7	5.5	1.2	5.2	3.2	1.6	16.1	3.8	2.4	87.4	102.4	89.5

Legend: CV, coefficient of variation; LOD, limit of detection; LLOQ, low limit of quantification; QC, quality control samples.

## Data Availability

The data presented in this study were obtained from the included studies and are openly available.

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
