# Peer review of "Development and Validation of a Fast and Sensitive UPLC-MS/MS Method for Ethyl Glucuronide (EtG) in Hair, Application to Real Cases and Comparison with Carbohydrate-Deficient Transferrin (CDT) in Serum"

_ijms, 2025, doi:10.3390/ijms26031344_

Round 1
Reviewer 1 Report
Comments and Suggestions for Authors
The paper provides a valuable contribution to forensic toxicology by presenting a fully validated method using UPLC-MS/MS for detecting ethyl glucuronide (EtG) in hair samples. The study's strengths include the adherence to established international guidelines (SoHT) and the innovative application of the M3® reagent, which significantly reduces sample preparation time. The research is timely and relevant, particularly given the increasing demand for reliable biomarkers in assessing chronic alcohol consumption in both clinical and forensic contexts.
The introduction provides a strong background and relevant references. Consider elaborating on the differences between the current method and previous studies for more context. This will strengthen the justification for the research.
The research design is robust and suitable for the objectives. It is suggested to clarify the rationale behind selecting the M3® reagent over other potential reagents to enhance transparency.
The methods are detailed and adhere to international guidelines. Please consider including a flowchart or table summarizing the workflow for improved accessibility, especially for multi-step processes.
Results are well-presented with relevant tables and figures. Further discuss the statistical significance of the findings, especially regarding the lack of concordance between hEtG and %CDT.
The conclusions are supported by the results. It is recommended to highlight more explicitly how the findings could influence clinical and forensic practices, emphasizing the implications for high-throughput analysis.
Author Response
Dear Reviewer,
in attach you will find the answers to your observations.
Thank for your comments.

Reviewer 2 Report
Comments and Suggestions for Authors
The manuscript entitled “Development and validation of a fast and sensitive UPLC-MS/MS method for ethyl glucuronide (EtG) in hair, application to real cases and comparison with Carbohydrate Deficient Transferrin (CDT) in serum” describes a method for detection of alcohol abuse through analysis of ethyl glucuronide (EtG) in hair and compare the result to CDT method. The following points should be considered before accepting the paper:
1- According to the attached similarity report, there is a high level of similarity (42%) which is unacceptable. Authors should rephrase their article to lower this high level.
2- Rationale of the study: Authors need to justify their selection of Ethyl glucuronide (EtG) as a biomarker for detection of alcohol abuse although there are many others that can be used.
3- Discussion: “One of the aims of our study was to compare two biomarkers of alcohol consumption”. Put a table in the discussion part to conclude the comparison between them and add some important items like: Cost-effectiveness, reliability, accuracy, availability and all other required items to give the reader a clear comparison.
4- Methodology: Statistical analysis method isn’t included in the methodology part. Write a short paragraph explaining it.
5- The citing method throughout the whole manuscript is incorrect. The reference number should be put within two brackets [ ].
6- References are written with the wrong format. Return to authors instructions to follow the correct format.

Author Response

(The authors gave the same response as above.)

Reviewer 3 Report
Comments and Suggestions for Authors
This article describes the validation of an LC-MS/MS method to detect ethyl glucuronide (EtG) in hair samples from drivers convicted of DUI. The authors combined recent hair processing results using M3 reagent test kit for sample processing with UHPLC-MS/MS for detection demonstrating higher throughput analysis vs processing hair samples via boiling in water. They also compared the results for EtG detection with another alcohol biomarker, serum CDG from the same pool of subjects, though the results showed no correlation between those biomarkers.
This article is well written and the experiments are, generally, adequately described to justify the conclusions including appropriate citations. The presentation is average with little data/figures in the manuscript.
Recommendation to editor: Accept with major revisions as detailed below
Minor Concerns:
Line 35: “Fully” change to “Full”
Figure 1: Label the Ethyl sulphate in the pathway like ethyl glucuronide
Line 103: “Fully” change to “Full”
Line 201: How long between driver conviction and sampling? Is this typically hours or weeks after charges? This is important missing information to support the claim of mismatching biomarker results between hEtG and CDT.
Line 202: “LMC” not defined.
Line 236: “blood” change to “serum”
Line 239: Volume injected? How were the injections concentration normalized between real samples?
Line 242: Please provide the identity of both mobile phases as well as the gradient details.
Line 248: I believe it should be “monosialo”
Line 259: “M1 and M2” please provide the specific identity of both mobile phases
Table 1: Swap the columns of M2 and M1 (M1 / Pump A should be first)
Line 275: “was” changed to “were”
Line 277: “prepared by diluting”
Line 277: what were the deuterium samples diluted with?
Line 278: change to “… solutions were prepared daily at…”
Line 282: was the EtG-free hair collected from alcohol abstinent subjects?
Line 297: please provide the identity of the precolumn
Line 304: “until to” change to “ by”
Line 306: “ at least one” changed to “one”
Table 2: Change both headings “Qualifier” to “Transition”. Also, the authors do not comment on the retention-time shift of the deuterated EtG. These should be eluting from the columns at the same retention time. In spiked samples they should co-elute. Could the authors explain and add a comment to the MS.
Table 3: “LOQ” change to “LLOQ”. Remove the QC label from each of the headings and put in the master heading above. The 2 in r2 needs to be superscript.
Major Concerns:
Line 299: More parameters are required for the MS method –source voltages, gas flows, and temperatures, dwell times, etc. It should be enough information for others to exactly recreate the method. What software was the data collected in? What software was the data processed in? How were the data processed?
Author Response

(The authors gave the same response as above.)

Round 2
Reviewer 2 Report
Comments and Suggestions for Authors
The authors addressed all the comments, and I approve this manuscript for publication.
Author Response
Thank you for your collaboration.
Kind regards
Reviewer 3 Report
Comments and Suggestions for Authors
1. RE the mobile phases being undisclosed because they are patented in both instances of the manuscript:
The authors should at least provide the specific name / model ID of Eureka lab kit they are using. I think it is (CDT TEST IN SERUM BY UV / VIS- FAST - MONOREAGENT – Code Z68215) from what I can find online. The reference provided [37] is not complete enough to find the exact source.
2. Why were details on the chromatography of CDT analysis removed from the revision? I understand the authors have referenced a previous report, were there no changes to the workflow? I believe the authors should reinclude at least the chromatography setup in the current manuscript (columns, system, temperature, etc.).
3. RE the author response: "The required data are not useful for reproducing the method because they are strictly dependent on the instrumentation used for analysis.":
I fundamentally disagree with the authors on this statement. It is imperative that all parameters of the experiment are provided. That these are specific only for their instrumentation is not justification for omitting them. They do not operate a unique system. Voltages, gas flows, and temperatures and all other MS parameters are essential for a complete and accurate LC MS methods section.
Author Response
Dear Reviewer attached are the answers to your questions.
